# Multidisciplinary team healthcare professionals' perceptions of current and optimal acute rehabilitation, a hip fracture example A UK qualitative interview study informed by the Theoretical Domains Framework

**Stefanny Guerra**[1], **Kate Lambe**[1], **Gergana Manolova**[1], **Euan Sadler**[2], **Katie J. Sheehan**[1]*

1 Department of Population Health Sciences, School of Life Course and Population Sciences, King's College London, London, United Kingdom, 2 Faculty of Environmental and Life Sciences, School of Health Sciences, University of Southampton, and Southern Health NHS Foundation Trust, Southampton, United Kingdom

* Katie.sheehan@kcl.ac.uk

**Data Availability Statement:** The authors did not receive consent from participants to share or

## Abstract

### Objective

To understand multidisciplinary team healthcare professionals' perceptions of current and optimal provision of acute rehabilitation, perceived facilitators and barriers to implementation, and their implications for patient recovery, using hip fracture as an example.

### Methods

A qualitative design was adopted using semi-structured telephone interviews with 20 members of the acute multidisciplinary healthcare team (occupational therapists, physiotherapists, physicians, nurses) working on orthopaedic wards at 15 different hospitals across the UK. Interviews were audio-recorded, transcribed verbatim, anonymised, and then thematically analysed drawing on the Theoretical Domains Framework to enhance our understanding of the findings.

### Results

We identified four themes: *conceptualising a model of rehabilitative practice*, which reflected the perceived variability of rehabilitation models, along with facilitators and common patient and organisational barriers for optimal rehabilitation; *competing professional and organisational goals*, which highlighted the reported incompatibility between organisational goals and person-centred care shaping rehabilitation practices, particularly for more vulnerable patients; *engaging teams in collaborative practice*, which related to the expressed need to work well with all members of the multidisciplinary team to achieve the same person-centred goals and share rehabilitation practices; and *engaging patients and their carers*, highlighting the importance of their involvement to achieve a holistic and collaborative approach to rehabilitation in the acute setting. Barriers and facilitators within themes were underpinned by

archive their personal or anonymized data publicly. Participants agreed to share their anonymized data for dissemination purposes, such as in journal publications and conferences. Therefore, excerpts of the transcripts relevant to our findings are available within the paper (in the text of the results and Table 2).

**Funding:** This work was supported by a UKRI Future Leaders Fellowship [Grant Ref: MR/S032819/1]. The funders had no role in study design, data collection and analysis, decision to publish, or preparation of the manuscript. https://www.ukri.org/what-we-offer/developing-people-and-skills/future-leaders-fellowships/.

**Competing interests:** UKRI Future Leaders Fellowship funding [Grant Ref: MR/S032819/1] provides salary support for KS and SG. KS is the Chair of the Scientific and Publications Committee of the Falls and Fragility Fracture Audit Programme which managed the National Hip Fracture Database audit at the Royal College of Physicians. ES is supported by NIHR ARC Wessex. GM and KL declare no competing interests. This does not alter our adherence to PLOS ONE policies on sharing data and materials (though there are restrictions in place for data sharing, as participants did not consent to share or archive their anonymised dataset publicly).

the lack or presence of adequate ways of communicating with patients, carers, and multidisciplinary team members; resources (e.g. equipment, staffing, group classes), and support from people in leadership positions such as management and senior staff.

## Conclusions

Cornerstones of optimal acute rehabilitation are effective communication and collaborative practices between the multidisciplinary team, patients and carers. Supportive management and leadership are central to optimise these processes. Organisational constraints are the most commonly perceived barrier to delivering effective rehabilitation in hospital settings, which exacerbate silo working and limited patient engagement.

## Introduction

Rehabilitation is defined as "a set of interventions designed to optimise functioning and reduce disability in individuals with health conditions in interaction with their environment" [1]. When delivered effectively, rehabilitation leads to improved patient, healthcare, and societal outcomes including reduction in health inequalities [2]. In the United Kingdom (UK), there is a translation gap between what is known to be effective and what is possible given available resources [3]. This translation gap leads to variation in the organisation of rehabilitation across care settings with commissioners making different decisions on how best to allocate available resources locally, regionally, and nationally [4, 5]. These decisions have knock on effects for clinical managers and clinicians themselves when determining how best to prioritise rehabilitation caseloads given available resources [6].

The first phase of rehabilitation reflects the onset of an acute illness or injury (or exacerbation of a chronic illness) often for patients with complex care needs requiring specialist support and predominantly takes place in the acute hospital setting [2]. This phase of rehabilitation would appear to be the most 'protocolised' as patients are cared for 24 hours a day 7 days per week in a standard hospital setting often with targeted key performance indicators to enable discharge as early as possible [7]. It is usual to anticipate a degree of variation in access to, and delivery of, rehabilitation interventions as individuals (even with the same diagnosis) will have different needs, abilities, and expectations for recovery [8]. However, the extent to which this variation is attributable to differences in patient characteristics has been called into question, with several reports of variation due to differences in the organisation and delivery of rehabilitation even during this initial more protocolised phase [9–12]. This variation has potentially negative implications for patients as of how well an individual progresses during this early phase of rehabilitation is often used as a criterion for access to further rehabilitative services across the care continuum [7].

Hip fracture is a good example of observed variations in acute rehabilitation despite a protocolised approach to care [13]. On average, 65,000 older adults are admitted with hip fracture to an acute hospital in England and Wales each year [14]. The injury reflects a heterogeneous population of older adults, many of whom present with other comorbidities, live in domiciliary and residential/nursing care settings, with different levels of prefracture functional ability and available social support [9, 15]. On admission to hospital, patients will begin a protocol for hip fracture care typically comprising six key performance indicators–prompt orthogeriatric assessment, prompt surgery, guideline recommended surgical approach, prompt mobilisation after surgery, assessment for delirium, and return to original residence, which are audited and

publicly reported nationally [14]. These indicators underpin a multidisciplinary team approach to care which is often dominated by rehabilitation during the acute hospital stay as most patients undergo surgery within 36-hours of an average stay of 15 days [14]. However, despite national audit and public reporting, variations in access and delivery of care persists for this patient population [16].

To date, there have been several qualitative studies exploring healthcare professional perspectives of variation in access to, and delivery of, rehabilitation after hip fracture [17–24]. These studies have mainly focused on individual professional groups [17, 24] and highlight resource constraints [17, 18, 22, 24], poor patient engagement [17, 19, 22–24], and limited multidisciplinary team engagement [17, 18, 20, 21, 23] as key contributors of unwarranted variation in rehabilitative care across hospitals. Despite the multidisciplinary nature of rehabilitation there are few studies which consider the perspectives of different multidisciplinary team members regarding what optimal rehabilitation after hip fracture looks like, and the perceived barriers to its implementation [20–22]. The Theoretical Domains Framework (TDF) offers a useful lens to explore this further as it was originally designed to identify determinants of current and desired behaviour that can lead to implementation problems, such as the delivery of optimal rehabilitation after hip fracture [25]. The TDF encompasses 12 domains: knowledge, skills, social/professional role and identity, beliefs about capabilities, beliefs about consequences, motivation and goals, memory attention and decision processes, environmental context and resources, social influences, emotion, behavioural regulation, and nature of behaviour/s. The domains enable structuring of qualitative data to identify behaviours and implementation barriers and facilitators to target for intervention. Once these determinants of behaviour are identified, they offer a useful framework for the choice of future quality improvement interventions.

The aim of this study was to understand multidisciplinary team healthcare professionals' perceptions of current and optimal provision of acute rehabilitation, perceived facilitators and barriers to implementation, and their implications for patient recovery, using hip fracture as an example. The analysis draws on the TDF to enhance our understanding of what professional behaviours and implementation facilitators and barriers to target, in order to improve provision of optimal rehabilitation in acute hospital settings.

## Materials and methods

This study is reported according to the Consolidated Criteria for Reporting Qualitative Research (COREQ) checklist" [26]. We received institutional ethical (REC reference: LRM-20/21-21197) and local governance approvals to conduct this study from the Research Ethics Office at Kings College London.

### Study design

A qualitative design was used to provide an in-depth understanding of multidisciplinary healthcare professionals' perspectives of current and optimal acute rehabilitation and perceived implementation facilitators and barriers. The study was underpinned by an interpretivist philosophical view of the social world which is based on the premise that our knowledge of reality is socially constructed by our perceptions and interpretations of it.

### Eligibility criteria

We aimed to recruit multidisciplinary team healthcare professionals, including physiotherapists, occupational therapists, nurses, and physicians with at least 2 years experience of working within acute rehabilitation after hip fracture in the UK. There was no additional inclusion

or exclusion criteria. This was in order to gain insight from a range of different professional groups.

## Sampling and recruitment

We used a convenience sampling approach [27] to recruit multidisciplinary team healthcare professionals by advertising the study through relevant professional societies (Chartered Society of Physiotherapy, Royal College of Occupational Therapists, Royal College of Nursing, and the British Geriatrics Society) and via Twitter.

## Data collection

Potential participants contacted one member of the research team (KL) by email to express their interest in taking part in the study, receive the participant information sheet and consent form, and ask questions. Interested participants return signed consent forms by email. Individual semi-structured telephone interviews were conducted by one author (KL). KL initially piloted the topic guide with one healthcare professional through established contacts with the research team after which the transcript was reviewed by three authors (KL, ES, KS) and the interview topic guide was further refined prior to commencing the interviews. The topic guide comprised a series of semi-structured open-ended questions and relevant prompts, when needed, seeking to capture multidisciplinary team healthcare professional perspectives on current and optimal provision of rehabilitation after hip fracture in an acute hospital setting, perceived barriers and facilitators to implementation, and implications for recovery. The topic guide was theoretically informed, with questions and prompts mapped to the TDF to ensure the topic guide would enable generation of data related to individual, social, and environmental determinants of behaviours and implementation barriers as part of this framework (S1 Table). Interviews were audio-recorded, transcribed verbatim and anonymised by an external professional translation service prior to data analysis.

## Data analysis

Data analysis proceeded until data saturation was deemed to have been reached, in which no new relevant themes were emerging from the qualitative data [28]. A thematic analysis approach was used to analyse and organise themes grounded in the qualitative data [29], drawing on the TDF [25] to enhance our understanding of what behaviours and implementation barriers and facilitators were perceived to influence optimal rehabilitation in acute hospital settings.

   Specifically, the qualitative analysis process involved a number of phases. The first phase involved three authors (SG, GM, KL) reading all transcripts, generating initial themes (codes), and grouping similar themes together (initial and axial coding) in NVivo (version 12) [29]. In the second phase these clusters of codes were used to organise initial themes into conceptual themes and related subthemes using the 'one sheet of paper method' approach, whereby similar and diverse perspectives among participants were identified across different professional groups [30]. The final phase involved mapping the findings within each theme to the TDF domains to identify behaviours and implementation barriers and facilitators perceived to influence optimal rehabilitation in acute hospital settings (see S1 Appendix for an example). These themes were refined iteratively with discussions within the research group. The final themes were discussed and agreed among the research team. A summary of final themes were also sent back to study participants by email for member checking, of whom only one participant replied stating the findings made sense to them.

**Research team and reflexivity.** All interviews were completed by KL a research assistant and health psychologist with prior experience of interviewing patients and healthcare professionals working with older adults with dementia. Participants were aware of KL's research role and that she did not have direct involvement with patient care. KL did not disclose any assumptions or reasons for doing the research and/or interest in the research topic prior to, during, or after conducting the interviews. Analyses were completed by SG, GM, and KL with iterative discussions with ES and KS. SG and GM are research assistants with experience of qualitative research. KS is a physiotherapist and researcher with expertise in hip fracture health services research. ES is a social scientist and physiotherapist working in social science applied health and implementation science research, with expertise in qualitative research methods. Thus, we considered the interdisciplinary nature of the research team enhanced quality in this study because the team brought together multiple perspectives to understand how acute rehabilitation after hip fracture could be optimised based on multidisciplinary team healthcare professionals' perceptions. This aligned with our interpretivist philosophical view of reality as socially constructed.

## Results

### Participant characteristics

Interviews (ranging between 32–51 minutes) were carried out with 20 health care professionals with a median of 17 years (interquartile range: 7, 21) of clinical experience (see Table 1). These included seven occupational therapists, six physiotherapists, three nurses, three geriatricians, and one orthopaedic surgeon, employed across England and Scotland. Most participants were female (n = 18) and had no research experience (n = 15).

### Themes

Four key themes and related subthemes were identified during the analysis: conceptualising a model of rehabilitative practice; competing professional and organisational goals; engaging teams in collaborative rehabilitation and; engaging patients and their carers. These themes were mapped to belief statements and domains of the TDF, with illustrative participant quotations in Table 2, and subsequently organised into perceived facilitators and barriers to implementation of optimal provision of rehabilitation in Table 3. Specific domains related to the TDF are indicated in brackets in the themes below.

**Conceptualising a model of rehabilitative practice.** This theme encompassed the perceptions of participants regarding the model of rehabilitation promoted by their service. Rehabilitation as described by participants varied, suggesting inconsistent protocolised approaches across sites despite similar hospital settings and established key performance indicators. Despite variations in the descriptions of rehabilitation practices, most healthcare professionals affirmed the specific practices of which they were part was working for their setting (*n = 16*). Participants also described a number of facilitators and barriers to implementing their perceived optimal model of rehabilitation.

Across services, recurring factors perceived to facilitate optimal rehabilitation (by at least 3 participants) included: teams working well together and supportive consultants and senior management who encouraged improvements to current rehabilitation services (*Social Influences*, *Social/professional role and identity*), organisational systems for patient notes and to prompt assessments, access to specialised professionals or services (e.g. orthogeriatricians, dieticians, specialised wards), having responsibility over patients' rehabilitation journey (e.g. deciding on referral pathway or discharge criteria), or providing activities to engage patients in rehabilitation (*Memory, attention and processes*, *Environmental context and resources*, *Belief about capabilities*, *Belief about consequences*):

**Table 1. Participant characteristics.**

| Participant ID | Gender | Occupation | Clinical experience (years) | Research experience | Location | Number of hip fractures seen per year at site |
|---|---|---|---|---|---|---|
| 1 | Male | Orthopaedic surgeon, lead clinician | 32 | Yes | South England | >300–500 |
| 2 | Female | Orthogeriatric consultant | 27 | Yes | North England | 100–300 |
| 3 | Female | Occupational therapist | 25 | No | North England | >300–500 |
| 4 | Female | Nurse | 23 | Yes | East England | 100–300 |
| 5 | Female | Orthopaedic physiotherapist, team lead | 21 | No | South England | >300–500 |
| 6 | Female | Orthopaedic physiotherapist, team lead | 21 | No | Scotland | >300–500 |
| 7 | Female | Nurse | 19 | No | Scotland | >500 |
| 8 | Male | Occupational therapist, team lead | 18 | No | East England | 100–300 |
| 9 | Female | Orthogeriatric consultant | 18 | No | North England | >300–500 |
| 10 | Female | Occupational therapist and senior research fellow | 17 | Yes | East England | 100–300 |
| 11 | Female | Trauma and orthopaedic physiotherapist | 17 | No | East England | >300–500 |
| 12 | Female | Trauma and orthopaedic physiotherapist, team lead | 13 | No | South England | >300–500 |
| 13 | Female | Occupational therapist | 12 | Yes | North England | >300–500 |
| 14 | Female | Physiotherapist, inpatient team lead | 10 | No | South England | >300–500 |
| 15 | Female | Trauma and orthopaedic physiotherapist | 7 | No | North England | <100 |
| 16 | Female | Occupational therapist | 7 | No | Scotland | >500 |
| 17 | Female | Occupational therapist, clinical lead | 7 | No | North England | >300–500 |
| 18 | Female | Occupational therapist | 6.5 | No | Scotland | >500 |
| 19 | Female | Orthogeriatric consultant | 3.5 | No | South England | >300–500 |
| 20 | Female | Nurse and clinical educator | 3.5 | No | East England | 100–300 |

**Table 2. Domains of the Theoretical Domains Framework as they relate to themes and belief statements, with supporting quotes from multidisciplinary participants.**

| Domain | Theme | Belief statements | Illustrative quotations |
|---|---|---|---|
| Knowledge | engaging teams in collaborative practice | communicating and learning from other health professionals helps to deliver optimal rehabilitation | "We meet with them [orthopeadic trauma group] every morning for a brief handover and then we're, we're kind of constantly in touch through the day really, it's a really great close working relationship where I can ask them about you know, why does somebody faint when they stand up or whether their pain in inhibiting their therapy or, and they can come and ask me because we're around on the ward a lot, we work very, very closely together." (P2, orthogeriatric consultant) |
|  | engaging patients and their carers | carers can provide helpful information about how to engage patients in rehabilitation | "We try and liaise with our carers and the relatives as often as possible, discuss any potential problems we might have with them such as how to, if they've become low in mood if it's normal for them and how, if there's anything they do at home to improve it." (P20, nurse) |

*(Continued)*

**Table 2.** (*Continued*)

| Domain | Theme | Belief statements | Illustrative quotations |
|---|---|---|---|
| **Skills** | competing professional and organisational goals | rehabilitation requires to regularly adapt to the constant variability of patients presenting with hip fracture | "You can have a very active 70-year-old. I would say all the patients are very different and very individual . . . I very much react to the patient, it's very individual, I react to the patient's needs at the time. I can't say I use a standard approach." (P3, OT) |
| | engaging teams in collaborative practice | healthcare professionals need to adapt their way of working together according to patients' individual needs and abilities | "So for somebody who is normally very well or functional, drives a car, gets out and about and they've literally tripped over something and broken a hip, then their rehabilitation is largely going to be the physiotherapist because their needs, otherwise, aren't so great. For somebody who is much frailer with cognitive impairment and delirium and lives at home and has a lot of functional deficit, then actually the physiotherapist may not have as much a role to play. It may be more occupational therapy and me and the nursing staff." (P9, orthogeriatric consultant) |
| | | multidisciplinary training and shared learning facilitate a standard and collaborative approach to rehabilitation | "So we've given the empowerment, if you like, we don't have to get a patient up on day zero, nursing staff will do it. So we've gone in with them, we've taught them, we've given them the competencies, they're competent to do it, they take the same assessments as we do and they can get them up and get them going." (P5, physiotherapist) |
| | engaging patients and their carers | healthcare professionals need to educate carers and patients on what optimal rehabilitation involves | "I think that health professionals have a role in talking to the family, quite often family can be overprotective and can wrap their loved ones up in cotton wool and it's about educating them as well in terms of being safe but encouraging activity or encouraging appropriate tasks to aid them in their recovery." (P10, OT and research fellow) |
| | | healthcare professionals need to support and reassure patients | "I think just thinking about the emotional bit as well, it's quite often people with a fractured hip experience trauma and that's quite often very emotional for them and they don't often see that straightaway . . . so it's just being ready for when that happens and being able to support them." (P10, OT and research fellow) |
| **Social/ professional role and identity** | conceptualizing a model of rehabilitative practice | supportive leadership and management are a main driver to deliver and improve rehabilitation | "I think we've got the right people behind us that have the drive not only to push these models forward but also to keep them going." (P20, nurse) |
| | competing professional and organisational goals | the imperative for early discharge is not always aligned with healthcare professionals' views of optimal rehabilitation | "[T]here's a big push to get people home and do all the care, the acute rehab in the home, but you know, I've always argued that a patient has to be able to do a basic minimum before they can get home." (P6, physiotherapist) |
| | | rehabilitation for more vulnerable patients is particularly challenging when aiming to meet organisational goals | "If you've got somebody who is incredibly elderly and frail who wasn't great before they came in and struggling, they're going to really struggle . . . pushing the physiotherapy two, three times a day." (P7, nurse) |
| | engaging teams in collaborative practice | collaborative practices are an essential aspect of rehabilitation | "I think it's really important that it's a multi-professional approach. I don't think one particular professional input is more valid than another, it really is a team effort with the end goal of getting the patient really to the best place on discharge." (P8, Band 7 OT) |
| | | collaborative practices are possible and maintained through supportive management and shared leadership | "I think it's just that ethos and that culture, and maybe between the senior sister and ourselves as team leaders within the therapy, whether that would perhaps help, if we had a bit more cohesion between us, that we'd then pass on throughout the teams." (P12, physiotherapist) |
| | engaging patients and their carers | explaining rehabilitation likely processes and reassuring patients is part of rehabilitation | "So with the patient, it's managing their expectations. You know, it's a big, catastrophic event for them so it's more a case of sort of explaining to them, this is fine, you will recover from this, education, education, education. This is what we expect you to get back to and this is how long it's going to take." (P5, physiotherapist) |
| | | all health professionals need to encourage patients taking ownership of their own recovery | "[I] explain to patients, part of your rehab isn't just the time that you spend with the physio or with the OT, it's also the time walking out to the bathroom with the nurse or the healthcare assistant or even by yourself is a part of your rehab because that's you starting to use your muscles again and starting to practice your walking etc, that lots of activity that you're doing in hospital without maybe another person being there with you." (P19, orthogeriatric consultant) |

(*Continued*)

**Table 2.** (*Continued*)

| Domain | Theme | Belief statements | Illustrative quotations |
|---|---|---|---|
| **Belief about capabilities** | conceptualizing a model of rehabilitative practice | optimal rehabilitation is facilitated when health professionals take responsibility and decide as a team over patients' journey | "We had very good MDT working, very good communication between particularly the OTs, the physios and the nurses on which patients we were accepting in the first place so, you know, on their referral we could say, yes they're absolutely appropriate, yes they've got rehab goals, yes this is for them." (P14, physiotherapist) |
| | engaging teams in collaborative practice | rehabilitation requires healthcare professionals working well together towards the same goals | 'I think it needs to be everybody working towards the same thing and if I's not then it's not going to work because it's you know, we can't do the physio rehab without the pain management or without the fluid management or without the skin care you know, everything's got to link up. "(P6, physiotherapist) |
| | | collaborative practices are facilitated when healthcare professionals are flexible about the perceived boundaries of their roles | "So physio and OT here tend to work quite separately, we don't tend to work as a big team, we tend to do a lot of separate working so physio you know, will go and see a patient in the morning and then OT will go later on but we don't tend to join up necessarily and sometimes I think there is a lot of duplication, so I think possibly if we could make a difference to maybe more joint working between physio and OT and seeing all patients with assistance of two that would help." (P18, OT) |
| | engaging patients and their carers | all health professionals need to encourage patients' independence | "It's not for us to start washing somebody that's washed themselves for seventy-odd years unless they actually need us to do it. So everything like that, promoting independence as much as possible, yeah, it's almost cruel to be kind. It's the more you do for somebody the less they're going to do and the less they're going to progress in rehab for you." (P7, nurse) |
| **Optimism** | conceptualizing a model of rehabilitative practice | rehabilitation becomes more challenging when patients have additional comorbidities or are from out of area | "[In] my very short career I've seen a massive change in the patients' presentation, their ability and their sort of like functional decline really" (P17, OT) |
| | competing professional and organisational goals | healthcare professionals need various adaptations to their typical way of working to rehabilitate more vulnerable patients | "We very much will still see them [patients with cognitive impairment] and try and make it functional, we'll try and work more with the nursing staff so you know, if the nursing staff are doing a wash and then the patient needs to toilet, so maybe we'll use that an opportunity to assess them transferring to get to the toilet." (P6, physiotherapist) |
| | | | "A lot of people obviously with cognitive impairment won't be able to work with you, so it's really trying to maximise what they can do, but they're not going to be able to engage with physiotherapy in a traditional sense of following instructions. So it's working out for each individual patient, as a team, what their goal of treatment and therapy is going to be." (P9, orthogeriatric consultant) |
| | engaging patients and their carers | additional activities and resources help ameliorate the emotional impacts of rehabilitation | "We have activities coordinator on the ward, and kind of if the patient's confused, you'll try and engage them in just like a small task, for example playing music and chatting with them . . . because sometimes to get a patient out into the chair and just, they'll just sit there, so he was very good at lifting patients' spirit. And he's quite vital to that patient journey. (P15, physiotherapist) |
| **Intentions** | conceptualizing a model of rehabilitative practice | sharing responsibilities helps to deliver rehabilitation in the face of organisational constraints | "A lot of our occupational therapy time can be documentation as well, doing referrals for packages of care, and community services and things like that, so you know, sometimes our physio colleagues will try and share the workload, which is also a great factor as well. And in turn, then you know, we've got quicker kind of assessments on the ward." (P16, OT) |
| | competing professional and organisational goals | advocating for patients as a team helps rehabilitation in the face of organisational constraints | "I think we've got enough people on the ward who are advocates for the patients, that we can normally get the result we want if we're facing adversity from kind of the powers that be, or from a discharge planning kind of aspect." (P11, physiotherapist) |
| | | working towards meeting organisational goals detracts from delivering person-centred care | "[I]n the acute service it's so driven towards just getting someone out of hospital that you can sometimes lose sight of that individual needs." (P16, physiotherapist) |

(*Continued*)

**Table 2.** (Continued)

| Domain | Theme | Belief statements | Illustrative quotations |
|---|---|---|---|
| **Goals** | competing professional and organisational goals | healthcare professionals' and organisational views of rehabilitation tend to differ | "Yes we can get somebody back to their care home within 3 days, and then the hospital management are happy and the NHS as a whole are happy because it's then a bed that we've freed up, but for that individual patient I don't think it's added very much to their care." (P9, orthogeriatric consultant) |
| | engaging patients and their carers | rehabilitation is facilitated by motivated patients taking ownership of their own recovery | "I think the patient has to subscribe and be up to participating in rehab otherwise it's just going to be a kind of a non-starter really. So it really does need for me the patient to be on board with anything that's going to happen in terms of rehabilitation to get them home." (P8, nurse) |
| **Beliefs about consequences** | conceptualizing a model of rehabilitative practice | providing additional activities and resources motivates patients and helps engage them with rehabilitation | "Little things such as having music on in the day really helps to uplift their spirits which then had a knock-on effect in improving their physio outcome, so little things like having a radio on has a positive impact. (P20, nurse and clinical educator) |
| | competing professional and organisational goals | working towards organisational goals is particularly detrimental for more vulnerable patients | "I feel very uncomfortable about the drive to get people out of hospitals back to care homes without giving them more time in rehabilitation. And I think getting back to the care homes is the entirely appropriate thing to do from a medical point of view, but then they get very little physiotherapy after they've gone back and I do worry that we're kind of consigning these people who are the most vulnerable patients that we have to additional dependence that they didn't have before." (P9, orthogeriatric consultant) |
| | engaging patients and their carers | lack of carers engagement is detrimental for optimal rehabilitation and not always reliable source of support, particularly for more vulnerable patients | "I think that's been, probably the biggest challenge since Covid in the fact that we can't get visitors in as freely, because I think, especially with some of our cognitively impaired patients, having a family member or a carer that they know well with them can have a massive impact on us being able to successfully rehab them" (P11, physiotherapist) |
| | | carers and patients' expectations impede recovery when not educated on what optimal rehabilitation involves | "I think sometimes patients perceptions of what rehab is, is very different to what it actually is . . . I mean you're literally talking about getting up and probably walking to the toilet or doing your bed or your chair exercises . . .. they're not going to do anything more than what they did before you know." (P17, OT) |
| | | in the acute setting, better outcomes are achieved when patients and carers take ownership of rehabilitation | "Often that [discharge] might be only three/four days, sometimes it can obviously take a lot longer, but then a patient wouldn't be anywhere near being fully recovered or rehabilitated in three/four/five days, so then it becomes reliant predominantly on the patient and the family themselves to rehabilitate them." (P13, OT) |
| **Memory, attention and decision processes** | conceptualizing a model of rehabilitative practice | frequent discussions and organisational systems for patients notes help guide rehabilitation practices | "It [organisational system] just contains everything and it prompts . . . so it's really just an easy way of overseeing the patient's journey basically from like a multidisciplinary point of view . . . helps you to sort of identify if the cognitive problems that people have got are new, and if they are you can highlight it and discuss with the MDT and ask them to assess it further." (P13, OT) |
| | engaging teams in collaborative practice | written communication helps to guide health professionals' roles to work towards the same goals as a team | "We also have a communication board with what their functional ability is on that day, how they're mobilising and how they're transferring. And then the nursing staff on that ward will follow that advice and continue with the patient, for example when the patients get back in bed or they want to go to the toilet. We very much see the rehab role as an MDT really." (P3, OT) |

(*Continued*)

**Table 2.** (Continued)

| Domain | Theme | Belief statements | Illustrative quotations |
|---|---|---|---|
| **Environmental context and resources** | conceptualizing a model of rehabilitative practice | specialised professionals and services facilitate rehabilitation | "I think we've got a really good service to be honest, and I think part and parcel of that is the fact that we have specified rehabilitation unit, it really helps get our flow, and also we're a funded service, so we're very well supported managerially, and so you know, when we need equipment or we need help with discharge planning, I think we're well supported from that point of view." (P15, physiotherapist) |
| | engaging teams in collaborative practice engaging patients and their carers | improving post discharge care helps overarching rehabilitation goals in the acute setting fluctuation of financial resources and staffing, are main impediments to delivering optimal rehabilitation | "I think if there was one thing I could make better it would probably to have more communication with the therapy teams who are looking after our patients once they leave our wards." (P2, orthogeriatric consultant) |
| | | | "I think if we're well staffed we can meet you know, and certainly and do pretty well with the audit and see people quickly, but I think as soon as we're pressured certainly over the winter months it can be really difficult and if we don't have the staff often it doesn't become as high a priority as people that are actually needing to go home that day." (P18, OT) |
| | | healthcare professionals are better able to engage in collaborative practices in the absence of organisational constraints | "I think because it's gotten much busier and probably a lot more for them [nursing staff] to do, I think they [nursing staff] just often find it easier just to [go] in and you know, quickly wash somebody rather than actually maybe spending the time with somebody saying you know, can you do this for yourself." (P18, OT) |
| | | | "Time constraints is huge, you tend to find your hip fracture patients need a lot of care and in the acute trauma wards it's just a very busy environment." (P9, orthogeriatric consultant) |
| | | the hospital environment in itself is a challenge for rehabilitation, particularly for those with cognitive impairment the covid-19 pandemic worsened patients' outcomes as it limited carers involvement, activities and resources to motivate/engage patients | "A lot of these patients are very cognitively impaired which obviously is a challenge and you put them in a single side room . . .They can't even recognise that they're in a hospital until the nurse comes in and tells them." (P7, nurse) |
| | | | "On the ward as well physio-wise they do, they've not been able to at the moment with Covid again but they'd started to do group therapy which was quite good and patients were quite willing to get involved and quite enjoyed that." (P4, nurse) |
| **Social Influences** | conceptualizing a model of rehabilitative practice | rehabilitation is strengthened when healthcare professionals are motivated, work well together, and support each other | Our therapists are brilliant, we've got a really great bunch who are, I think they're pretty well led and they're pretty focussed on what they're doing and they're really interested in getting things better. . . I guess being part of a team like this is really great and enormously encouraging and uplifting. (P2, orthogeriatric consultant) |
| | engaging teams in collaborative practice | health care professionals need to support and trust each other to deliver rehabilitation collaboratively | "It is about respecting the skills that each profession can give . . . which will benefit the patient longer term." (P9 OT) |
| **Behavioural regulation** | conceptualizing a model of rehabilitative practice | planning and communicating and planning with patients as soon as possible facilitates rehabilitation | "If you speak to people from the minute they come in they've got an idea of the pathway and how it's going to progress over the next, well, for the duration of their inpatient stay so it gives them something to think about and kind of work towards. So yeah, I think communication's probably the easiest way to improve it. (P7, nurse) |
| | engaging teams in collaborative practice | peer-feedback supports others to manage actions through audit or informal processes to extend skills | "We do a lot of reflective practice. . . a lot of in-service training. Anyone that's come up against a new piece of equipment we'll make half an hour to go through it. Anyone that has had a difficult conversation with a family member, okay, how did you deal with that? what did you do? let's do that next time, let's not do that. If it's a difficult conversation on the phone can someone listen in, can anyone provide any help?" (P17, OT) |

"*I think our model works well because we're as a team we're quite interested in improving care, not that other teams aren't, but we're just really enthusiastic and we're quite eager. I think we've got the right people behind us that have the drive not only to push these models forward but also to keep them going.*" (P20, female, nurse and clinical educator, 3.5 years of experience)

**Table 3. Summary of multidisciplinary team perceived barriers and facilitators to acute rehabilitation service delivery after hip fracture according to domains of the Theoretical Domains Framework.**

| Associated Theoretical Domain | Facilitators |
|---|---|
| **Knowledge** | Engaging with carers and patients as soon as possible to obtain information needed to deliver person-centred care, especially for more vulnerable patients |
| | Effective and frequent communication amongst health professionals to discuss patients optimal care and learn from each other |
| **Skills** | Training within and across disciplines involved in rehabilitation to better work collaboratively |
| | Educating patient and carers on best practices for rehabilitation, and to manage expectations |
| **Social/professional role and identity** | Supportive management and leadership supporting and providing healthcare professionals with the flexibility required to provide person-centred care |
| | Supportive management and leadership supporting improvement and development practices |
| **Beliefs about capabilities** | Healthcare professionals being able to decide on patients' rehabilitation journey to deliver person-centred care |
| | Healthcare professionals sharing responsibilities to ensure ongoing rehabilitation, whilst also supporting patients' independence and ownership of rehabilitation |
| **Optimism; and Environmental context and resources** | Providing additional activities to engage and improve patients' mood, and reinforce a positive attitude towards rehabilitation |
| | Adapting rehabilitation for more vulnerable patients |
| **Belief about consequences** | Patients taking ownership for their own recovery |
| | Communicating with patients and carers as soon as possible and throughout hospital stay to address concerns and reassure them |
| | Engaging carers in rehabilitation, especially with more vulnerable patients |
| **Intentions** | Sharing rehabilitation amongst health professionals, or advocating for patients as a team, to mitigate organisational constraints |
| **Memory, attention and decision processes** | Organisational systems and frequent meetings that remind and inform healthcare professionals of patients' assessments, rehabilitation goals and medical care |
| **Social influences; and Social/professional role and identity** | A positive culture where all healthcare professionals communicate and work well together, respecting and learning from each other |
| | Supportive management and shared leadership that encourages and promotes this positive culture |
| **Behavioural regulation** | Monitoring progress and identifying areas for improvement |
| | **Barriers** |
| **Social/professional role and identity; and Belief about capabilities** | Healthcare professionals' belief of their role in rehabilitation, characterised by distinct priorities and a reluctancy to step in other professionals' role |
| **Belief about capabilities** | Healthcare professionals not working collaboratively to supporting patients' independence |
| | Healthcare professionals not deciding over patients' rehabilitation journey |
| **Belief about consequences** | Lack of patients and carers engagement, or overprotective carers |
| | Patients and carers holding unrealistic expectations of rehabilitation |
| | Patients with cognitive impairment who cannot take ownership for their own rehabilitation |
| **Optimism; and Environmental context and resources** | Patients out of area and erratic linkages to community care |
| | Patients presenting with additional comorbidities |
| **Intentions; and Social/professional role and identity** | Prioritisation of patients to meet organisational goals that do not match healthcare professionals view of optimal rehabilitation and person-centred care |

*(Continued)*

**Table 3.** (Continued)

| Associated Theoretical Domain | Facilitators |
|---|---|
| **Goals; and Environmental context and resources** | Organisational goals of reducing length of hospital stay not aligned with professionals' goals of delivering person-centred care, particularly for the more vulnerable patients |
| **Environmental context and resources** | Shortages, fluctuation of resources |
| | Lack of carers engagement and stopping additional activities to engage patients in rehabilitation, due to the covid-19 pandemic |

Less frequently reported facilitators of optimal rehabilitation (reported by at least 1–2 participants) included establishing a therapeutic relationship with a healthcare professional, early communication and planning with patients, or strengthening post discharge care (e.g., follow patients up for outreach work or to gather feedback, links with community rehabilitation) (*Belief about consequences, Behavioural regulation, Environmental context and resources*). For example, one occupational therapist said:

> "*Doing the split post with acute and community gives me the opportunity to . . .. give advice and education to the staff on the acute ward in terms of how to improve rehabilitation in the acute setting to help the more longer-term rehabilitation*" (P13, female, occupational therapist, 12 years of experience)

Where individual participants thought rehabilitation fell below expectations, this often related to organisational changes shaping the rehabilitation service in hospital, or a shortage and fluctuation of resources such as financial provisions and staffing (*Environmental context and resources*). There were various perceived causes for these shortages, for example, financial constraints in funding more staff positions; disruptions due to the covid-19 pandemic; difficulties in recruitment; getting cover for seven-day service and for staff leave. Participants from different professional groups shared the view that they were left dissatisfied and aware they were not providing the perceived optimal rehabilitation for patients.

> "*I think the model's okay; I just wish we had more of it.*" (P1, male, lead clinician orthopaedic surgeon, 32 years of experience)

> "*I don't know anywhere that's genuinely delivering seven days, a seven-day orthogeriatric service, I'm absolutely certain you can't do it with two consultants.*" (P2, female, orthogeriatric consultant, 27 years of experience)

The impact of organisational issues (including staff shortages) was mitigated when healthcare professionals worked closely together to deliver shared rehabilitation practices (*Intentions*). This shared practice was considered to maximise opportunities for rehabilitation while minimising unnecessary repetition of practice through crossing of professional boundaries. This approach was highlighted by physiotherapists, occupational therapists and nurses and most often implemented when rehabilitation was considered to encompass an array of care processes inclusive of but not limited to mobility e.g., discharge planning, activities of daily living including washing, dressing. For instance, one nurse commented:

> "*They might not have funding to get more physiotherapists, but they've changed the way they work . . . certainly it has improved over the last few years. They* [patients] *are not getting their*

*activity co-ordinator, their OT, and their physio all in one day and then sitting dormant for five or six days, so it's spread out during the week and then nursing staff are still doing rehab and walking people to the toilet."* (P7, female, nurse, 19 years of experience)

Some healthcare professionals expressed concerns over a perceived change in the extent to which care is patient centred, inhibiting optimal rehabilitation (*Optimism*). This shift was seen to be due to two factors–a changing clinical presentation of the population, and erratic linkages to community care. Health professionals highlighted patients are presenting with greater complexity due to multimorbidity and increased levels of dependency. This complexity was perceived to steer the focus towards planning for discharge which was not always person-centred (as some patients would benefit from more rehabilitation during the acute stay). Perceived erratic community linkages led to uncertainties over reliability of referrals following discharge and a lack of confidence in relaying to patients what they should expect from their ongoing rehabilitation (and a desire to retain in the acute setting to optimise recovery).

*"I think for example because we're a tertiary service we get patients out of area, and I think that sometimes can be a barrier within itself when it comes to discharge planning, because we can't give them the same standard of care when it comes to going to rehabilitation"* (P15, female, trauma and orthopaedic physiotherapist, 7 years of experience)

*"In my very short career I've seen a massive change in the patients' presentation, their ability and their sort of like functional decline really"* (P17, female, clinical lead occupational therapist, 7 years of experience)

**Competing professional and organisational goals.**   Participants commonly commented on a mismatch between the flexibility required to adjust to individual needs (*Skills*) and the organisational goals for a standardised, pre-set model for rehabilitation after hip fracture (*Social/professional role and identity*). This was often reflected by healthcare professional goals of a good foundation for functional recovery on discharge, and organisational goals for discharge home as soon as possible (*Goals*). These competing goals sparked frustrations with participants emphasising the challenges of making a one-size-fits-all model work for the diverse scope of patients that they see with hip fracture (*Intentions*):

*"There's a big push to get people home and do all the care, the acute rehab in the home, but you know, I've always argued that a patient has to be able to do a basic minimum before they can get home."* (P6, female, team lead orthopaedic physiotherapist, 21 years of experience)

*"Any models are set up for the majority, not for the individual patient, despite everyone aiming to be patient-centred."* (P9, female, orthogeriatric consultant, 18 years of experience)

The majority of participants highlighted that more vulnerable patients, including those from care homes and/or with cognitive impairment, were deemed to be more negatively affected by organisational drivers for early hospital discharge. Indeed, such participants emphasised that higher numbers of patients admitted from care homes were discharged with worse outcomes that those admitted from their own home setting, whilst higher numbers of patients with cognitive impairment transitioned to care homes than those without cognitive impairment (*Beliefs about consequences*). These poorer patient outcomes were attributed to an organisational imperative to quickly discharge patients and subsequent prioritisation based on anticipated potential (*Social/professional role and identity*):

"*I feel very uncomfortable about the drive to get people out of hospitals back to care homes without giving them more time in rehabilitation. And I think getting back to the care homes is the entirely appropriate thing to do from a medical point of view, but then they get very little physiotherapy after they've gone back, and I do worry that we're kind of consigning these people who are the most vulnerable patients that we have to additional dependence that they didn't have before.*" (P9, female, orthogeriatric consultant, 18 years of experience)

To mitigate the negative impact on more vulnerable patients, half of participants (n = 10) spoke about making adaptations to care, such as asking family to join for rehabilitation, expediting discharge to return patients to familiar surroundings, placing patients in enhanced care bays, or holding dedicated recreational activities. Although there was limited discussion of the outcomes of these strategies, such health professionals focus was on gaining patients' trust and making them feel comfortable, working with family members, and adapting sessions to patients needs and abilities (*Environmental context and resources*, *Optimism*). A few participants (n = 6) also advocated for healthcare professionals shifting away from organisational goals, although this was influenced by team dynamics such as how well teams communicate, listen and respect each other opinions, the degree of support and flexibility enabled by management (*Intentions*). For example, one physiotherapist commented:

"*I think we've got enough people on the ward who are advocates for the patients, that we can normally get the result we want if we're facing adversity from kind of the powers that be, or from a discharge planning kind of aspect.*" (P11, female, trauma and orthopaedic physiotherapist, 17 years of experience)

**Engaging teams in collaborative practice.**   This theme reflected participants' perceptions regarding their relationships with other healthcare professionals and the expressed need to work collaboratively to maximise recovery and likelihood of returning home following rehabilitation for patients with hip fracture. This collaborative practice was facilitated by a positive team culture, underpinned by communication, appropriate resources, and supportive leadership and management.

For most (n = 17), the collaborative nature of their work was underscored in the discussion of their own role and others' perceived role in rehabilitation (*Professional role and identity*). Participants often commented on perceived unique and overlapping areas of their professional practice and how the engagement of each health professional may vary depending on the needs of an individual patient (*Skills*), for instance one consultant said:

"*For somebody who is normally very well or functional, drives a car, gets out and about and they've literally tripped over something and broken a hip, then their rehabilitation is largely going to be the physiotherapist because their needs, otherwise, aren't so great. For somebody who is much frailer with cognitive impairment and delirium and lives at home and has a lot of functional deficit, then actually the physiotherapist may not have as much a role to play. It may be more occupational therapy and me and the nursing staff.*" (P9, female, orthogeriatric consultant, 18 years of experience)

Implementing this collaborative way of working was closely related to professional perceptions of a positive team culture, commonly defined by cooperation, a smooth handover between healthcare professionals, learning from each other on the job and freely voicing one's professional opinion. As such, successful collaboration was underpinned by a mutual respect

and support across areas of practice. This was achieved via the shared leadership among senior staff who instilled the cooperative atmosphere and actively promoted it to other professionals and new members of the team (*Social influences, Social/Professional role and identity*):

> "*We don't do one thing without asking the other first. . . for example I wouldn't mobilise my patient without checking with my physio . . .what stages they are at with regards to their mobility. I don't make assumptions on where my patient's going without speaking to the orthogeriatricians.*" (P20, female, nurse and clinical educator, 3.5 years of experience)

The main feature underpinning a positive team culture was considered to be good communication facilitated by multidisciplinary team meetings, bedside whiteboards, systems for organizing staff notes, and/or clinical governance meetings. This intentionally or implicitly worked to align the attitudes, values, and care outlook among healthcare professionals. Over half of participants (n = 16) reported they engaged in almost daily multidisciplinary team meetings or ward rounds where each professional commented on a patient's management from their own professional perspective. This was an opportunity to reinforce positive team dynamics and parity, for professionals to learn from each other, broaden their care perspective, and modify their approach to accommodate this broader perspective (*Knowledge; memory, attention, and decision processes*):

> "*We also have a communication board with what their functional ability is on that day, how they're mobilising and how they're transferring. And then the nursing staff on that ward will follow that advice and continue with the patient, for example when the patients get back in bed or they want to go to the toilet. We very much see the rehab role as an MDT* [multidisciplinary team] *really. The nurses are very focussed on also trying to improve someone's mobility.*" (P3, female, occupational therapist, 25 years of experience)

A positive team culture was also enabled by dedicating time to support shared learning within and across professional groups *(Skills)*. This learning included both formal (in-service training) and informal (support to extend skills) training which was sometimes evaluated through e.g., audit to enable advocacy for additional resource (*Behavioural regulation*), but often not, as one OT voiced:

> "*We've given the empowerment, if you like, we don't have to get a patient up on day zero, nursing staff will do it. So we've gone in with them, we've taught them, we've given them the competencies, they're competent to do it, they take the same assessments as we do and they can get them up and get them going.*" (P5, female, team lead orthopaedic physiotherapist, 21 years of experience)

Several participants (*n = 7)* spoke about aspiring to these ways of working which more effectively blurred perceived boundaries of professional roles to ensure that care and rehabilitation was provided irrespective of issues with staffing (*Beliefs about capabilities*):

> "*I think if it was a whole team approach of promoting independence it'd be much more helpful, I think it's really difficult, like I've had a few instances this week where you know, we're going in ourselves and the physio and say you know, you need to be doing these on your own and then nursing staff are coming along and saying oh you know, we'll wheel you to the toilet, it's just really not helpful*" (P18, female, occupational therapist, 6.5 years of experience)

Implementing this in practice, however, was considered challenging, due to organisational constraints such as shortages and heavy workloads, or professionals' perceptions and priorities for their unique role in rehabilitation (*Environmental context and resources*, *Belief about capabilities*). This viewpoint was typically illustrated by the occupational therapist below:

"*I think, traditionally the focus is more on, oh, okay, well physiotherapists get people walking, occupational therapists deal with equipment, the nursing deal with continence and nutrition, sort of nursing care and continence and things like that. Whereas, actually, I think for everything to work there is an overlap between that. And to get the best sort of model of care is where you can all truly work together and have sort of a fluid role, in some respects, between the different professions.*" (P12, female, team lead trauma and orthopaedic physiotherapist, 13 years of experience)

**Engaging patients and their carers.** This theme related to professional attitudes towards working with patients and their families during early rehabilitation. Patient engagement with rehabilitation and the degree of involvement of relatives and carers were considered leading factors for successful rehabilitation in hospital.

All participants perceived they adopted a person-centred approach to rehabilitation with a commonly shared belief voiced that improved outcomes were achieved when patients take ownership of their own recovery. Promoting this positive attitude in individual towards rehabilitation was considered of particular importance in the context of limited physiotherapist and/or occupational therapy staff resources (*Belief about consequences*, *Goals*). To reinforce this individual responsibility of the patient, health care professionals often felt they needed to present a unified front to support patients' independence, by reminding and facilitating this approach to rehabilitation among different team members (*Belief about capabilities*, *Social/professional role and identity*). For instance, one consultant commented:

"*I explain to patients, part of your rehab isn't just the time that you spend with the physio or with the OT, it's also the time walking out to the bathroom with the nurse or the healthcare assistant or even by yourself is a part of your rehab because that's you starting to use your muscles again and starting to practice your walking etc, that lots of activity that you're doing in hospital without maybe another person being there with you.*" (P19, female, orthogeriatric consultant, 3.5 years of experience)

All healthcare professionals acknowledged that taking ownership for their early rehabilitation after hip fracture would not be possible for all patients. In particular, the challenge of supporting patients with cognitive impairment to engage in rehabilitation was identified across all professional groups (*Belief about consequences*). A number of professionals regarding such patients commented, "[they] don't fall in line with the model" but also acknowledged that there was no alternate model of rehabilitation for these complex patients. Some indicated "the responsibility is placed upon the people who work with them, the carers, the family to encourage any kind of rehabilitation". Others acknowledged that care is delivered "opportunistically" and can vary considerably from one professional and patient/carer-dyad to the next.

Different healthcare professionals also acknowledged that such an approach to rehabilitation may be challenging for the older patient population presenting with hip fracture in an acute care setting, due to factors such as frailty, comorbidities, delirium, and/or disruptive and busy hospital environments. Participants across all professional groups (n = 5) found it helpful to provide additional activities (group therapy, music, volunteers, support to dress in own

clothes) to engage patients with rehabilitation and support a positive attitude. These however relied mainly on adequate resources and staff's extra time, and many of these additional activities had been stopped because of the Covid-19 pandemic *(Environmental context and resources, Optimism)*:

> *On the ward as well physio-wise they do, they've not been able to at the moment with Covid again but they'd started to do group therapy which was quite good and patients were quite willing to get involved and quite enjoyed that."* (P4, female, nurse and clinical educator, 23 years of experience)

Over half of participants (n = 12), representing different professional groups, described in detail typical interactions with patients following hip fracture, which commonly included explaining about care pathways, managing expectations, encouraging progress, and supporting a positive individual attitude towards recovery (*Social/professional role and identity*). Several (n = 6) also highlighted the importance of directly acknowledging the emotional burden presented by hip fracture and subsequent need for supported rehabilitation to address both the physical and psychological aspects of recovery (*Skills*). For example, one physiotherapist said:

> "*With the patient, it's managing their expectations. You know, it's a big, catastrophic event for them so it's more a case of sort of explaining to them, this is fine, you will recover from this, education, education, education. This is what we expect you to get back to and this is how long it's going to take.*" (P5, female, team lead orthopaedic physiotherapist, 21 years of experience)

Several Healthcare professionals also highlighted the essential role of carers for successful rehabilitation. Communication with carers was perceived as paramount to obtain information about the patients' preferences and goals, particularly in the case of patients with cognitive impairment, to recruit them as reassuring and motivating presence during rehabilitation, and to arrange follow-up support after discharge. In the context of limited resources, carers engagement in rehabilitation were considered a key advantage (*Knowledge, Belief about consequences*). This belief was emphasised by most participants to be challenging during the Covid-19 pandemic where access to carers was limited to the perceived detriment of patients with hip fracture (*Environmental context and resource*).

> "*I think that's been, probably the biggest challenge since Covid in the fact that we can't get visitors in as freely, because I think, especially with some of our cognitively impaired patients, having a family member or a carer that they know well with them can have a massive impact on us being able to successfully rehab them*" (P11, female, trauma and orthopaedic physiotherapist, 17 years of experience)

Under pre-pandemic circumstances, different healthcare professionals perceived available support varied widely in part due to competing responsibilities of carers (e.g., work, and childcare commitments) and the feasibility of their support (*Belief about consequences*). Furthermore, certain family carers were sometimes a perceived barrier to patients' rehabilitation progress if they adopted an overprotective stance or had unrealistic expectations for progress for their relative. A number of health professionals thus felt they needed to educate carers on the likely milestones for rehabilitation (*Belief about consequences*) and how to encourage progress in line with best practice (*Skills*).

*"You get patients and their carers complaining, they say well, they made them toilet them-selves or they watched them do this and they didn't provide care for them, they just observed them or assisted them in doing an activity and they don't seem to understand that that's the whole point of it is for us to enable them to re-enable and rehab, so we now emphasise like the whole point of this is for them to improve their skills and not for us to do stuff with them because they will start to lose their ability to do this and that's not what the aim of this is, the aim is to get them back to near as their baseline function as much as possible."* (P19, female, orthogeriatric consultant, 3.5 years of experience)

## Discussion

### Main findings

This study focused on multidisciplinary team healthcare professionals' perceptions on current and optimal provision of acute rehabilitation, perceived facilitators and barriers to implementation, and their implications for patient recovery using hip fracture as an example population. Four key themes were identified during the analysis: *conceptualising a model of rehabilitative practice, competing professional and organisational goals, engaging teams in collaborative reha-bilitation, and engaging patients and carers*. Themes were interpreted through the lens of the TDF to identify perceived behaviours and implementation facilitators and barriers to target for intervention.

In accordance with reported sources of variation, we found that the main determinants of optimal rehabilitation were organisational features [17, 18, 23, 24] and engagement of patients, carers and the multidisciplinary team [17–24]. For these to be addressed, the presence of sup-portive management and leadership often stood out as essential to promote a positive culture where multidisciplinary teams, adequately trained and supported, communicated and worked well together towards person-centred goals. Services worked towards these ideals in distinct ways, in line with the variations in care provision found for hip fracture rehabilitation [16] and the contextual variability evident across individual hospitals when implementing services [31].

### Facilitators of optimal rehabilitation

Communication was perceived by healthcare professionals as the central implementation facil-itator of optimal provision of rehabilitation. This communication was noted at several levels–with the patient and carer, among healthcare professionals, and with senior management and leadership. Key features included 1) timing -early engagement of all healthcare professionals, patients and carers to ensure appropriate understanding of prefracture capability (*Knowledge*), common expectations for rehabilitation (*Skills*), and optimize engagement (*Optimism, Envi-ronmental context and resource, Belief about consequences*), and 2) frequent communication -particularly among healthcare professionals to ensure close monitoring of progress (*Knowl-edge*), shared learning (*Social influences, Social/professional role and identity*) across disciplines, and consistent information and practices with patients. Such early, frequent, and holistic approach to communication is supported by Health Education England's recommendation for effective multidisciplinary teams working in health care [32], as long as it is also used to better establish and deliver person-centred care. The newly proposed key performance indicator 'zero' (assessing pain relief and admission to an appropriate ward within 4 hours of presenting with a hip fracture) [16] represents an opportunity for acute rehabilitation services to work towards this early engagement and potentially improve patient and multidisciplinary team engagement.

Shared responsibility for rehabilitation (*Intention*) was also identified as a facilitator of optimal provision of rehabilitation with multidisciplinary team training (*Social influences*, *Social/ professional role and identity*, *Skills*) to equip all members of the team (including patients and carers) to deliver key components of rehabilitation irrespective of professional background. The desire for other health professionals to aid with therapy and share opportunities to rehabilitate hip fracture patients has been expressed in other studies [17, 19]. Previous research has shown that patient benefits arise when nurses incorporate rehabilitation practices into their work [33]. These collaborative practices, however, can be perceived as intrusive to others professional roles, unrealistic in the face of heavy workloads and shortages, and may be hampered by professional tensions and lack of adequate training [32, 33]. These organisational constraints often exacerbate silo working among health professionals working in acute hospitals [34]. This way of working relies on environments that fosters a culture of collaboration, where multidisciplinary teams respect, listen and trust each other, feel valued, are appropriately trained, and have clarity over their responsibilities [32–35], a task that heavily lies on senior management and leadership [32–35].

Hence unsurprisingly emphasised was the importance of supportive management and shared leadership which stimulates communication through formal organisational structures such as meetings (*Memory*, *attention and decision processes*), monitors progress and areas for improvement (*Behavioural regulation*) and provides healthcare professionals flexibility to adapt provision enabling person-centred care (*Belief about capabilities*, *Social/professional role and identity*). Healthcare professionals have indicated elsewhere this facilitator as a main driver of effectively implementing services for hip fracture patients in the acute setting [18], and of promoting activities of daily living in hospitalised older adults [35].

## Barriers to optimal rehabilitation

A commonly perceived implementation barrier among healthcare professionals in this study was the limited patient and carer engagement, potentially due to complexity such as cognitive impairment, but which leads on to unrealistic expectations for rehabilitation (Belief about consequences). Particularly believed as detrimental for recovery outcomes was patients not taking ownership for their own rehabilitation journey. The importance of this responsibility has been priorly reported for this patient population [17, 23]. In accordance, this and other studies state health professional's role on informing, educating, and encouraging patients and carers [17, 19–24]. However, research suggests information and knowledge may not be enough for older patients to self-motivate when hospitalised if organisational goals, rather than person-centred goals, are the main focus of rehabilitation [36], as it often seems to be the case. Strategies previously described to encourage patients' engagement are the provision of alternative activities [20, 21, 24], communication skills training [19], goal setting [21], and booklets to remind key information and exercises [21]. We found that strategies to engage patients took the least priority and were inconsistent within and across settings, with most relying on staff's initiatives and extra time. These also tended to focus on patients with cognitive impairment, though patients who have broken their hip find it difficult to self-motivate regardless of cognitive status [23].

Healthcare professionals working in silos, focusing on distinct priorities, and a reluctance to step into other professionals' perceived roles (*Social/professional role and identity*, *Belief about capabilities*), was another main perceived barrier which aligns with previous studies [19, 22, 24]. Lack of coordination between multidisciplinary team members is related to delays in mobilisation [34], which in turns relates to worse recovery and survival outcomes [37]. Strategies previously identified to engage multidisciplinary teams working with hip fracture patients

are training, effective communication, and visual reminders [17, 18, 20, 23, 35]. We also observed vast variability in the way these strategies were implemented. There are nevertheless potential patient benefits from blurring professionals' boundaries. A nurse-led orthogeriatric care program for patients with hip fracture showed reductions in mortality by 3 and 12 months in comparison to usual care [38], and involved joint work from geriatricians, surgeons, physiotherapists, and occupational therapists to share rehabilitation responsibilities and learning.

Organisational characteristics, most commonly voiced as an implementation barrier by healthcare professionals, included protocol constraints limiting the need for flexibility to enable patients centred care (*Belief about capabilities*), shortages and/or fluctuation of resources, erratic links to community care limiting effective discharge planning (*Environmental context and resources*), and limitations of the acute hospital environment with insufficient resource to allow home visits during the hospital stay (*Environmental context and resources*). The new proposed Key Performance Indicator 7 (follow patients up 120 days post discharge to check on bone strengthening medication) [16] is an opportunity to improve referral pathways and linkages with community services, a crucial gap repeatedly highlighted for rehabilitation in the acute setting [17, 23, 24].

Organisational protocols that impede person-centred care have also been reported in hip fracture rehabilitation [17, 19]. From a broader rehabilitation perspective, physiotherapists and occupational therapists talk about an ideal for their practice (holistic improvements that return patients back to their pre-fracture functional status) that is inevitable unmet in the reality of the acute setting [39], a conflict attributed in large part to the priority of adhering to organisational standards [39]. Research describing the incompatibility of hip fracture rehabilitation models for hospitalised patients with dementia [24] and those in a less severe state [23], also deem organisational barriers that result in prioritisation of patients based on rehabilitation potential as a main contributor [23, 24]. Here, services were at least partly guided by key performance indicators, which resulted in modifications to strengthen and improve services but were also a reinforcer to the push to meet organisational goals rather than deliver person-centred care, impacting more vulnerable patients to a greater extent. A systematic review evaluating the experiences of healthcare professionals with implementation in acute settings highlighted successful interventions had considered the individual culture and organisational barriers of each site [31]. Furthermore, interventions were less likely to be reported as successful if they were not aligned with established hospital standards, as professionals prioritised these [31]. In line with our findings, this suggests that optimal rehabilitation interventions need to carefully balance the importance of person-centred care and the need to meet organisational goals.

## Wider implications of study findings

The focus of the current study was on rehabilitation after hip fracture as an example. Key implementation facilitators shared among multidisciplinary team healthcare professionals were communication, shared responsibility for rehabilitation, and supportive management and shared leadership. Key implementation barriers included absence of patient and carer engagement, healthcare professionals working in silos, and organisational barriers. While we focused on hip fracture the facilitators (and mechanisms to implement) and barriers (and mechanisms to overcome) are likely similar across admitting diagnoses for older adults. This is evidenced by studies on implementation of stroke care guidelines [40, 41] mobility and functional decline for a variety of diagnoses in hospitalised older adults [34, 42] rehabilitation for critically ill patients [43], and a review of hospital-based interventions [31].

## Limitations

We employed a convenience sampling approach with 20 participants working in 15 hospitals in the UK. This may have led to overstating perceived barriers and/or facilitators as some participants were working at the same hospital. We sought to capture a multidisciplinary perspective on rehabilitation however, participation was dominated by physiotherapists and occupational therapists (n = 14), those with at least 10-years of experience (n = 14), and who were female (n = 18) despite efforts to recruit other professional groups from multiple sources. This may reflect a perception that rehabilitation is a therapist's role opposed to a care structure/process [17]. This may lead to an imbalance of the perspectives of healthcare professionals more broadly limiting generalisability of the findings. Future research may focus on underrepresented groups (in terms of profession, experience, and sex) to broaden our understanding of optimal acute rehabilitation from the perspective of more groups. Moreover, alternative sampling strategies, such as snowball sampling, a procedure commonly used to increase the sample diversity of studies among 'difficult-to-reach' populations [44] may complement future research recruitment strategies. Finally, the study captured participants working in England and Scotland and the results may not be translated more widely to other settings where e.g., length of acute hospital stay may vary.

## Conclusions

Optimal rehabilitation in the acute setting requires effective communication and involvement of multidisciplinary teams, patients, and carers, to engage in a collaborative model of rehabilitation where individuals work towards the same person-centred goals. This collaborative way of working can then also ameliorate some of the organisational constraints. However, at the same time, organisational barriers (e.g. lack of resources and the need to meet organisational standards) can exacerbate silo working and poor patient engagement. There is variability in the way acute rehabilitation services work to attain these aims, but important facilitators to implement optimal acute rehabilitation services after hip fracture are the provision of adequate resources and supportive management and leadership characteristics within multidisciplinary healthcare professional teams.

## Supporting information

**S1 Table. Questions and prompts mapped to the Theoretical Domains Framework domains and constructs.**
(DOCX)

**S1 Appendix. Audit trail of phases of qualitative data analysis process with examples.**
(DOCX)

## Acknowledgments

We would like to thank the participants who gave up their time to take part in the interviews.

## Author Contributions

**Conceptualization:** Katie J. Sheehan.

**Data curation:** Stefanny Guerra, Kate Lambe, Gergana Manolova.

**Formal analysis:** Stefanny Guerra, Gergana Manolova.

**Funding acquisition:** Katie J. Sheehan.

**Investigation:** Kate Lambe.

**Methodology:** Kate Lambe, Euan Sadler, Katie J. Sheehan.

**Project administration:** Stefanny Guerra, Kate Lambe, Katie J. Sheehan.

**Supervision:** Euan Sadler, Katie J. Sheehan.

**Visualization:** Stefanny Guerra, Katie J. Sheehan.

**Writing – original draft:** Stefanny Guerra, Gergana Manolova, Katie J. Sheehan.

**Writing – review & editing:** Stefanny Guerra, Kate Lambe, Euan Sadler, Katie J. Sheehan.

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
