## [Decision Letter · Decision Letter 0]

16 Aug 2022

PONE-D-22-09908

Multidisciplinary perspectives of current and optimal acute rehabilitation, a hip fracture example: A qualitative interview study informed by the Theoretical Domains Framework.

PLOS ONE

Dear Dr. Guerra,

Thank you for submitting your manuscript to PLOS ONE. After careful consideration, we feel that it has merit but does not fully meet PLOS ONE’s publication criteria as it currently stands. Therefore, we invite you to submit a revised version of the manuscript that addresses the points raised during the review process.

Please note that we have only been able to secure a single reviewer to assess your manuscript. We are issuing a decision on your manuscript at this point to prevent further delays in the evaluation of your manuscript. Please be aware that the editor who handles your revised manuscript might find it necessary to invite additional reviewers to assess this work once the revised manuscript is submitted. However, we will aim to proceed on the basis of this single review if possible. 

Please see the reviewer's comments below - I agree that you could provide further details about participant selection and recruitment.

We look forward to receiving your revised manuscript.

Kind regards,

Steve Zimmerman, PhD

Associate Editor, PLOS ONE

https://journals.plos.org/plosone/s/file?id=ba62/PLOSOne_formatting_sample_title_authors_affiliations.pdf".

“UKRI Future Leaders Fellowship funding [Grant Ref: MR/S032819/1] provides salary support for KS and SG. KS is the Chair of the Scientific and Publications Committee of the Falls and Fragility Fracture Audit Programme which managed the National Hip Fracture Database audit at the Royal College of Physicians. ES is supported by NIHR ARC Wessex. GM and KL declare no competing interests.”

Reviewers' comments:

Reviewer's Responses to Questions

**Comments to the Author**

1. Is the manuscript technically sound, and do the data support the conclusions?

Reviewer #1: Yes

2. Has the statistical analysis been performed appropriately and rigorously? 

Reviewer #1: N/A

3. Have the authors made all data underlying the findings in their manuscript fully available?

Reviewer #1: No

4. Is the manuscript presented in an intelligible fashion and written in standard English?

Reviewer #1: Yes

5. Review Comments to the Author

Reviewer #1: First, thank you for the work. The topic is much needed and the Theoretical Domains Framework is an excellent choice to study health care professionals perceptions of current and optimal provision of rehabilitation in the acute hospital setting. It would be useful to include proposed solutions to recruit nurses and physicians (i.e. snowball sampling through included participants). I am curious which efforts you have done to recruit other healthcare professionals and whether they informed you why they did not want to take part in the study (answers given in line with your proposed perception that rehabilitation is a therapist’s role?)

Additionally, I personally prefer the term person-centered above patient-centered emphasizing to look after a person instead of only the disease/injury which sounds also more in line with the engagement of patients and carers in rehabilitation.

Moreover, I wondered if you formulated in- and exclusion criteria for participating in this study. Table 1 can thereby be improved by adding experience of participants with rehabilitation of patients with hip fracturs (e.g. how many seen per year) next to clinical experience. Another small remark on Table 1: Would be nicer to read when it is sorted according to e.g. years of clinical experience.

Finally, only two participants were male. Did you have the intention to recruit a certain percentage/distribution of men/women, less vs high experienced professionals? If not, what was the underlying reason? And if yes, why did it not work out?

Thanks again for your work.

6. PLOS authors have the option to publish the peer review history of their article (what does this mean?). If published, this will include your full peer review and any attached files.

Reviewer #1: No

---

## [Author Response · Author response to Decision Letter 0]

13 Sep 2022

Thank you for requesting us to resubmit our revised manuscript. Please find below our answers to your queries and suggestions. 

EDITOR COMMENTS

We have edited and checked style requirements are met. 

2. Thank you for stating the following in the Competing Interests section: “UKRI Future Leaders Fellowship funding [Grant Ref: MR/S032819/1] provides salary support for KS and SG. KS is the Chair of the Scientific and Publications Committee of the Falls and Fragility Fracture Audit Programme which managed the National Hip Fracture Database audit at the Royal College of Physicians. ES is supported by NIHR ARC Wessex. GM and KL declare no competing interests.” 

Please confirm that this does not alter your adherence to all PLOS ONE policies on sharing data and materials, by including the following statement: ""This does not alter our adherence to PLOS ONE policies on sharing data and materials.” (as detailed online in our guide for authors http://journals.plos.org/plosone/s/competing-interests). If there are restrictions on sharing of data and/or materials, please state these. Please include your updated Competing Interests statement in your cover letter; we will change the online submission form on your behalf.

An updated Competing Interests statement is now included in the Cover letter.

3. In your Data Availability statement, you have not specified where the minimal data set underlying the results described in your manuscript can be found. PLOS defines a study's minimal data set as the underlying data used to reach the conclusions drawn in the manuscript and any additional data required to replicate the reported study findings in their entirety. 

There are restrictions to sharing our data publicly. We have expanded on the details of these restrictions in the Data Availability statement. 

Our Data Availability statement now reads: 

We did not receive consent from participants to share or archive their personal or anonymised data publicly. Participants agreed to share their anonymised data for dissemination purposes, such as in journal publications and conferences. Therefore, we have made excerpts of the transcripts relevant to our findings available within the paper (in the text of the results and Table 2). 

REVIEWERS COMMENTS

1. Have the authors made all data underlying the findings in their manuscript fully available? Reviewer #1: No

Our Data Availability statement now reads: 

We did not receive consent from participants to share or archive their personal or anonymised data publicly. Participants agreed to share their anonymised data for dissemination purposes, such as in journal publications and conferences. Therefore, we have made excerpts of the transcripts relevant to our findings available within the paper (in the text of the results and Table 2). 

. 

2. First, thank you for the work. The topic is much needed and the Theoretical Domains Framework is an excellent choice to study health care professionals perceptions of current and optimal provision of rehabilitation in the acute hospital setting. It would be useful to include proposed solutions to recruit nurses and physicians (i.e. snowball sampling through included participants). I am curious which efforts you have done to recruit other healthcare professionals and whether they informed you why they did not want to take part in the study (answers given in line with your proposed perception that rehabilitation is a therapist’s role?)

Thank you for agreeing with us on the importance of the topic and choice of framework. 

We updated the methods section to read: 

We intended to recruit an even proportion of physiotherapists, occupational therapists, nurses, and physicians with at least 2 years experience of working within acute rehabilitation after hip fracture in the UK, or until we reach data saturation (REF). This was to gain insight from a range of different professional groups. By using a convenience sampling approach [27], we recruited 15 hospital sites through relevant societies (Chartered Society of Physiotherapy, Royal College of Occupational Therapists, Royal College of Nursing, and the British Geriatrics Society) and via social media. Potential participants contacted one member of the research team (KL) to express their interest in taking part in the study, receive answers to their questions, and provide informed written consent prior to scheduling their interview. We limited recruitment of physiotherapists and occupational therapists at 14 as no new themes were emerging from these professional groups. 

We also updated the limitations section of the discussion to read: 

‘We sought to capture a multidisciplinary perspective on rehabilitation however, participation was dominated by physiotherapists and occupational therapists (n =14), those with at least 10-years of experience (n = 14), and who were female (n = 18) despite efforts to recruit other professional groups from multiple sources. This may reflect a perception that rehabilitation is a therapist’s role opposed to a care structure/process[17].’ This may lead to an imbalance of the perspectives of healthcare professionals more broadly limiting generalisability of the findings. Future research may focus on under-represented groups (in terms of profession, experience, and sex) to broaden our understanding of optimal acute rehabilitation from the perspective of more groups. Moreover, alternative sampling strategies, such as snowball sampling, a procedure commonly used to increase the sample diversity of studies among ‘difficult-to-reach’ populations [REF to add– Kirchherr, 2018] may complement future research recruitment strategies.’

3. Additionally, I personally prefer the term person-centered above patient-centered emphasizing to look after a person instead of only the disease/injury which sounds also more in line with the engagement of patients and carers in rehabilitation.

We updated from patient-centered to person-centered throughout the manuscript.

4. Moreover, I wondered if you formulated in- and exclusion criteria for participating in this study. Table 1 can thereby be improved by adding experience of participants with rehabilitation of patients with hip fracturs (e.g. how many seen per year) next to clinical experience. Another small remark on Table 1: Would be nicer to read when it is sorted according to e.g. years of clinical experience.

Table 1 is now sorted by clinical years of experience. 

We added a row in Table 1 to indicate the number of hip fractures seen per year next to location. 

5. Finally, only two participants were male. Did you have the intention to recruit a certain percentage/distribution of men/women, less vs high experienced professionals? If not, what was the underlying reason? And if yes, why did it not work out?

We updated the methods section to specify:

‘We intended to recruit an even proportion of physiotherapists, occupational therapists, nurses, and physicians with at least 2 years experience of working within acute rehabilitation after hip fracture in the UK’

We did not sample based on sex and have updated the limitations section to read:

‘We sought to capture a multidisciplinary perspective on rehabilitation however, participation was dominated by physiotherapists and occupational therapists (n =14), those with at least 10-years of experience (n = 14), and who were female (n = 18) despite efforts to recruit other professional groups from multiple sources. This may reflect a perception that rehabilitation is a therapist’s role opposed to a care structure/process[17].’ This may lead to an imbalance of the perspectives of healthcare professionals more broadly limiting generalisability of the findings. Future research may focus on under-represented groups (in terms of profession, experience, and sex) to broaden our understanding of optimal acute rehabilitation from the perspective of more groups.’

---

## [Decision Letter · Decision Letter 1]

29 Sep 2022

PONE-D-22-09908R1Multidisciplinary perspectives of current and optimal acute rehabilitation, a hip fracture example: A qualitative interview study informed by the Theoretical Domains Framework.PLOS ONE

Dear Dr. Guerra,

Thank you for submitting your manuscript to PLOS ONE. After careful consideration, we feel that it has merit but does not fully meet PLOS ONE’s publication criteria as it currently stands. Therefore, we invite you to submit a revised version of the manuscript that addresses the points raised during the review process. Please consider the comments I have given below. 

Please submit your revised manuscript by Nov 13 2022 11:59PM. If you will need more time than this to complete your revisions, please reply to this message or contact the journal office at plosone@plos.org. Please include the following items when submitting your revised manuscript:A rebuttal letter that responds to each point raised by the academic editor and reviewer(s). You should upload this letter as a separate file labeled 'Response to Reviewers'.A marked-up copy of your manuscript that highlights changes made to the original version. You should upload this as a separate file labeled 'Revised Manuscript with Track Changes'.An unmarked version of your revised paper without tracked changes. You should upload this as a separate file labeled 'Manuscript'.If applicable, we recommend that you deposit your laboratory protocols in protocols.io to enhance the reproducibility of your results. Protocols.io assigns your protocol its own identifier (DOI) so that it can be cited independently in the future. For instructions see: https://journals.plos.org/plosone/s/submission-guidelines#loc-laboratory-protocols. Additionally, PLOS ONE offers an option for publishing peer-reviewed Lab Protocol articles, which describe protocols hosted on protocols.io. Read more information on sharing protocols at https://plos.org/protocols?utm_medium=editorial-email&utm_source=authorletters&utm_campaign=protocols.

We look forward to receiving your revised manuscript.

Kind regards,

Andrew Soundy

Academic Editor

PLOS ONE

Journal Requirements:

Additional Editor Comments:

Abstract

Calling it a qualitative semi-structured interview study – can change please identify the methodology

Methods

You again call it a qualitative study drawing on the framework – fine but what is the methodology and what is your world view? The framework presumably acted to inform your interviews and structure your analysis this is where the detail is needed.

Can you confirm the eligibility criteria was just at least 2 years experience? Was there no exclusion criteria?

Please separate a section on sampling and sample size from eligibility criteria. Also may be consider putting line 119 start around contacting participants into data collection section

Remove [28] from line 124 – not sure why it is there

Please add an audit trail for the reader to understand what you did – for instance we need a reference for your inductive thematic analysis which suggests an initial process of open coding? Correct? Why not use a framework analysis. Can you be clear on what you mean by aid interpretation of findings – you mean you used it as a framework for major themes then identified minor themes? You say initial themes – to be clear you mean themes from open coding or from the framework? Please talk about phases and reference a supplementary file for examples of each phase. When you mention reference 31 again it is hard to follow what you have done.

You mention saturation again here – given how big the TDF is I am surprised that you have captured all domains but fine.

You identify information regarding reflexivity but need to talk about how you enhanced quality and what you consider as quality based on your world view

Reviewers' comments:

Reviewer's Responses to Questions

**Comments to the Author**

1. If the authors have adequately addressed your comments raised in a previous round of review and you feel that this manuscript is now acceptable for publication, you may indicate that here to bypass the “Comments to the Author” section, enter your conflict of interest statement in the “Confidential to Editor” section, and submit your "Accept" recommendation.

Reviewer #1: All comments have been addressed

2. Is the manuscript technically sound, and do the data support the conclusions?

Reviewer #1: Yes

3. Has the statistical analysis been performed appropriately and rigorously? 

Reviewer #1: N/A

4. Have the authors made all data underlying the findings in their manuscript fully available?

Reviewer #1: Yes

5. Is the manuscript presented in an intelligible fashion and written in standard English?

Reviewer #1: Yes

6. Review Comments to the Author

Reviewer #1: Happy to see that all comments have been addressed by the authors. I have no further comments. Thanks again for this work!

7. PLOS authors have the option to publish the peer review history of their article (what does this mean?). If published, this will include your full peer review and any attached files.

Reviewer #1: No

---

## [Author Response · Author response to Decision Letter 1]

22 Oct 2022

Thank you for taking the time to review our manuscript. We have revised the draft based on your comments (inclusive of a review of the consistency in terminology throughout) and believe it to be much improved. All changes are highlighted in yellow on the revised draft. 

1. Abstract: Calling it a qualitative semi-structured interview study – can change please identify the methodology 

We updated the abstract methods to read: 

‘A qualitative design was adopted using semi-structured telephone interviews with 20 members of the acute multidisciplinary healthcare team (occupational therapists, physiotherapists, physicians, nurses) working on orthopaedic wards at 15 different hospitals across the UK. Interviews were audio-recorded, transcribed verbatim, anonymised, and then thematically analysed drawing on the Theoretical Domains Framework to enhance our understanding of the findings.’ 

2. Methods: You again call it a qualitative study drawing on the framework – fine but what is the methodology and what is your world view? The framework presumably acted to inform your interviews and structure your analysis this is where the detail is needed. 

We replaced the following paragraph on the TDF from the Study design section to the Introduction: 

‘The domains enable structuring of qualitative data to identify behaviours and implementation barriers and facilitators to target for intervention. Once these determinants of behaviour are identified, they offer a useful framework for the choice of future quality improvement interventions.’ 

We also updated sections of the methods to include the following text: 

Study design 

‘A qualitative design was used to provide an in-depth understanding of multidisciplinary healthcare professionals’ perspectives of current and optimal acute rehabilitation and perceived implementation facilitators and barriers. The study was underpinned by an interpretivist philosophical view of the social world which is based on the premise that our knowledge of reality is socially constructed by our perceptions and interpretations of it.’ 

Data collection 

‘The topic guide was theoretically informed, with questions and prompts mapped to the TDF to ensure the topic guide would enable generation of data related to individual, social, and environmental determinants of behaviours and implementation barriers as part of this framework (Supplementary File 1).’ 

Data analysis 

‘Data analysis proceeded until data saturation was deemed to have been reached, in which no new relevant themes were emerging from the qualitative data [27]. A thematic analysis approach was used to analyse and organise themes grounded in the qualitative data [29], drawing on the TDF [25] to enhance our understanding of what behaviours and implementation barriers and facilitators were perceived to influence optimal rehabilitation in acute hospital settings. 

Specifically, the qualitative analysis process involved a number of phases. The first phase involved three authors (SG, GM, KL) reading all transcripts, generating initial themes (codes), and grouping similar themes together (initial and axial coding) in NVivo (version 12) [29]. In the second phase these clusters of codes were used to organise initial themes into conceptual themes and related subthemes using the ‘one sheet of paper method’ approach, whereby similar and diverse perspectives among participants were identified across different professional groups [30]. The final phase involved mapping the findings within each theme to the TDF domains to identify behaviours and implementation barriers and facilitators perceived to influence optimal rehabilitation in acute hospital settings (see Supplementary file 2 for an example). These themes were refined iteratively with discussions within the research group.’ 

3. Can you confirm the eligibility criteria was just at least 2 years experience? Was there no exclusion criteria? Please separate a section on sampling and sample size from eligibility criteria. Also may be consider putting line 119 start around contacting participants into data collection section 

We updated the Eligibility and Sampling sections to read: 

Eligibility criteria 

‘We aimed to recruit multidisciplinary team healthcare professionals, including physiotherapists, occupational therapists, nurses, and physicians with at least 2 years experience of working within acute rehabilitation after hip fracture in the UK [27]. There was no additional inclusion or exclusion criteria.’ This was in order to gain insight from a range of different professional groups.’ 

Sampling and recruitment 

‘We used a convenience sampling approach [28] to recruit multidisciplinary team healthcare professionals by advertising the study through relevant professional societies (Chartered Society of Physiotherapy, Royal College of Occupational Therapists, Royal College of Nursing, and the British Geriatrics Society) and via Twitter.’ 

Data collection 

‘Potential participants contacted one member of the research team (KL) by email to express their interest in taking part in the study, receive the participant information sheet and consent form, and ask questions. Interested participants return signed consent forms by email. ‘Individual semi-structured telephone interviews were conducted by one author (KL). [...]’ 

4. Remove [28] from line 124 – not sure why it is there 

Apologies, this was a mistake which we have now corrected. 

5. Please add an audit trail for the reader to understand what you did – for instance we need a reference for your inductive thematic analysis which suggests an initial process of open coding? Correct? Why not use a framework analysis. Can you be clear on what you mean by aid interpretation of findings – you mean you used it as a framework for major themes then identified minor themes? You say initial themes – to be clear you mean themes from open coding or from the framework? Please talk about phases and reference a supplementary file for examples of each phase. When you mention reference 31 again it is hard to follow what you have done. 

We updated the data analysis section. Please see the updated paragraph in the response to comment 2. 

We also added a supplementary file (S2 Appendix I) with examples of each phase of our analysis. 

6. You mention saturation again here – given how big the TDF is I am surprised that you have captured all domains but fine. 

We updated the data analysis section to read: 

‘Data analysis proceeded until data saturation was deemed to have been reached, in which no new relevant themes were emerging from the qualitative data [27]. A thematic analysis approach was used to analyse and organise themes grounded in the qualitative data [29], drawing on the TDF [25] to enhance our understanding of what behaviours and implementation barriers and facilitators were perceived to influence optimal rehabilitation in acute hospital settings.’ 

7. You identify information regarding reflexivity but need to talk about how you enhanced quality and what you consider as quality based on your world view. 

We updated the text to read: 

‘Thus, we considered the interdisciplinary nature of the research team enhanced quality in this study because the team brought together multiple perspectives to understand how acute rehabilitation after hip fracture could be optimised based on multidisciplinary team healthcare professionals’ perceptions. This aligned with our interpretivist philosophical view of reality as socially constructed.’

---

## [Editor Report · Decision Letter 2]

8 Nov 2022

Multidisciplinary team healthcare professionals’ perceptions of current and optimal acute rehabilitation, a hip fracture example: A UK qualitative interview study informed by the Theoretical Domains Framework

PONE-D-22-09908R2

Dear Dr. Guerra,

We’re pleased to inform you that your manuscript has been judged scientifically suitable for publication and will be formally accepted for publication once it meets all outstanding technical requirements.

Kind regards,

Andrew Soundy

Academic Editor

PLOS ONE

Additional Editor Comments (optional):

Thank you for your positive responses to the concerns raised.
---

## [Editor Report · Acceptance letter]

10 Nov 2022

PONE-D-22-09908R2 

Multidisciplinary team healthcare professionals’ perceptions of current and optimal acute rehabilitation, a hip fracture example
A UK qualitative interview study informed by the Theoretical Domains Framework 

Dear Dr. Guerra:

I'm pleased to inform you that your manuscript has been deemed suitable for publication in PLOS ONE. Congratulations! Your manuscript is now with our production department. 

Kind regards, 

on behalf of

Dr. Andrew Soundy 

Academic Editor

PLOS ONE